# Extreme-Low-Speed Heavy Load Bearing Fault Diagnosis by Using Improved RepVGG and Acoustic Emission Signals

**DOI:** 10.3390/s23073541

**Published:** 2023-03-28

**Authors:** Peng Jiang, Wenyu Sun, Wei Li, Hongyu Wang, Cong Liu

**Affiliations:** 1School of Mechanical Science and Engineering, Northeast Petroleum University, Daqing 163318, China; 2Beijing Tongtai Hengsheng Technology Co., Ltd., Beijing 100020, China

**Keywords:** intelligent fault diagnosis, deep learning, extreme-low-speed heavy load bearing, MFCCs, EMA, RepVGG, wind turbine

## Abstract

With the worldwide carbon neutralization boom, low-speed heavy load bearings have been widely used in the field of wind power. Bearing failure generates impulses when the rolling element passes the cracked surface of the bearing. Over the past decade, acoustic emission (AE) techniques have been used to detect failure signals. However, the high sampling rates of AE signals make it difficult to design and extract fault features; thus, deep neural network-based approaches have been proposed. In this paper, we proposed an improved RepVGG bearing fault diagnosis technique. The normalized and noise-reduced bearing signals were first converted into Mel frequency cepstrum coefficients (MFCCs) and then inputted into the model. In addition, the exponential moving average method was used to optimize the model and improve its accuracy. Data were extracted from the test bench and wind turbine main shaft bearing. Four damage classes were studied experimentally. The experimental results demonstrated that the improved RepVGG model could be employed for classifying low-speed heavy load bearing states by using MFCCs. Furthermore, the effectiveness of the proposed model was assessed by performing comparisons with existing models.

## 1. Introduction

With changes in the world’s energy structures, installation capacity and investments in renewable energy have witnessed a stable and sustained growth. In July 2020, the EU announced the goal of achieving carbon neutrality by 2050 [1]. In September 2020, China proposed to peak carbon dioxide emissions by 2030 and to achieve carbon neutrality by 2060; in addition, China aims to achieve 1200 GW of wind power by 2030, which is equivalent to quadrupling the installation target for 2019 [2]. Therefore, new challenges are being put forth for the maintenance and fault detection of wind turbines.

Bearings are essential constituents of heavy rotating machinery; they mitigate the effects of friction between rotating shafts and stationary components, such as the bearing housing [3]. Bearing defects account for a significant portion (45–55%) of rotating machinery failures. Therefore, detecting incipient bearing faults at an early stage is crucial to preventing consequential failures of the manufacturing equipment [4]. In recent decades, many bearing fault diagnosis techniques have been developed based on acoustic emission (AE). AE-based analysis enables detection of very low-energy signals caused by bearing failures at an early stage or during low-speed operation [5,6,7]. However, because the sampling rate used for AE signal collection is usually greater than 1 MHz, analysis of AE signals is difficult because of the tremendous amount of data in the collected time series and the computational time required for analysis [8].

With the advent of computer technology and machine learning (ML), signal processing technologies, such as fast Fourier transform (FFT) [9], kurtosis spectrum analysis [10], envelope analysis [11], and variational mode decomposition (VMD) [12], have been developed for bearing fault diagnosis. In the field of ML, artificial intelligence (AI) algorithms, such as support vector machine (SVM) [13], random forest (RF) [14], and backpropagation neural network (BPNN) [15], have been developed. In 2006, Hinton proposed deep learning (DL) technology; subsequently, it was widely employed for machine fault diagnosis [16]. Grezmak et al. directly converted one-dimensional (1D) vibration signals into two-dimensional images and then used the convolutional neural network (CNN) to diagnose the images [17]. Guo et al. used the CNN to diagnose 1D vibration signals after Fourier transformation and achieved ideal classification results [18]. However, two problems are encountered with low-speed heavy load bearings: first, AI algorithms, such as SVM, do not fully utilize the powerful feature self-extraction ability of DL, and the extremely low rotation rate makes feature extraction challenging. Second, the working conditions of heavy-duty bearings are relatively bad, with a lot of noise and interference; as such, the fault signal is difficult to detect at low speeds.

This study primarily aimed to examine a novel dataset designed for detecting faults in low-speed heavy bearings used in industrial settings [19]. The commonly used CWRU dataset has been subject to various concerns, which were discussed in 2015 by Smith and Randall [20] and, more recently, in 2022 by Hendriks et al. [20,21]. These studies reported that the CWRU dataset may not accurately represent bearing faults in general and that it is even less representative of the specific industrial case analyzed in this study. Moreover, this study is motivated by the fact that although the use of CNNs for bearing fault diagnosis has been analyzed in the literature, CNNs are difficult to deploy in industry due to the requirement for a large number of model parameters. RepVGG’s structural reparameterization technique can greatly reduce the number of model parameters during reasoning, thus making it suitable for industrial applications [22].

In this study, a DL method was designed for AE signals for extreme-low-speed heavy load bearings for the use of large-sized bearings in industrial wind turbines. This is the first work that includes experiments conducted on large-size industrial bearings with localized faults. First, considering the low frequency of bearing fault signals at low speeds, the proposed method uses denoised Mel frequency cepstrum coefficients (MFCCs) as the input of the model. VMD-kurtosis denoising is employed as the denoising method to reduce high-frequency signals and to remove a large amount of background noise. Second, for industrial applications, the RepVGG convolutional network is employed for bearing fault diagnosis. The RepVGG network uses structure reparameterization technology to increase the reasoning speed of the model and to decrease the memory requirement, thereby enabling the remote deployment of the RepVGG for industrial applications. The experimental results corroborated this hypothesis and demonstrated that the feature extraction capabilities of the RepVGG network can be effectively utilized for fault diagnosis. Furthermore, for the same reasoning sample and approximate prediction performance, the reasoning speed of RepVGG is considerably better than that of other DL models.

This paper comprises five sections. In Section 1, the topic of intelligent fault diagnosis of industrial bearings is presented, and the authors’ motivations for undertaking this investigation in light of the existing literature are outlined. In Section 2, details regarding the AI methodologies utilized in this study, particularly VMD-kurtosis denoising MFCCs, CNNs, and RepVGG, are presented. In Section 3, the test rig employed for industrial bearings is described, and the results of a wind turbine bearing fault experiment conducted with AE signals are presented. In Section 4, the findings, discussions, and implications of the investigation are presented. Finally, the concluding remarks are provided in Section 5.

## 2. Proposed Method

In this section, a summary of the main method employed in this study is provided. VMD-kurtosis denoising, MFCCs, CNNs, and the RepVGG feature extractor are discussed.

### 2.1. VMD-Kurtosis Denoising

The VMD method treats the problem of mode decomposition as an optimization problem by decomposing the 1D input signal into a specified number of modes. The signal is fully reproduced by summing up the K number of decomposition modes:(1)$$H(t)=\sum_{k=1}^{K}u_{k}\left(t\right),$$
where k is the index of modes, K is the total number of modes, and $u_{k}\left(t\right)$ is the k-th mode and is an amplitude-modulated–frequency-modulated signal.

The decomposition of the time series by using the VMD method can be considered a constrained variational problem (Dragomiretskiy and Zosso, 2014) [23], whose objective function can be expressed as follows:(2)$$\left\{\begin{array}{c} \underset{\left\{u_{k}\right\},\left\{w_{k}\right\}}{\min}\left\{\sum_{k}\|\partial_{t} [\left(\delta \left(t\right)+\frac{j}{\pi t}\right){*u}_{k}\left(t\right)]e^{{-\mathrm{jw}}_{k}t}{\|}_{2}^{2}\right\}\\ s.t.\sum_{k}u_{k}=f \end{array}\right.,$$
where $\left\{u_{k}\right\}=\left\{u_{1},\ldots {,u}_{k}\right\}$ and $\left\{w_{k}{=w}_{1},\ldots {,w}_{k}\right\}$ are, respectively, the sets of all modes and their related center frequencies, $\delta \left(t\right)$ is the Dirac function, * denotes the convolution, and j = $\sqrt{-1}$.

Dragomiretskiy and Zosso (2014) used a quadratic penalty term and Lagrangian multiplier to transform Equation (2) into an unconstrained optimization problem:(3)$$L\left(\left\{u_{k}\right\},\left\{w_{k}\right\},\lambda \right)≔\alpha \sum_{k}\|\partial_{t\;} [\left(\delta \left(t\right)+\frac{j}{\pi t}\right){*u}_{k}\left(t\right)]e^{{-\mathrm{jw}}_{k}t}{\|}_{2}^{2}+\|f\left(t\right)-\sum_{k}u_{k}\left(t\right){\|}_{2}^{2}+\langle \lambda \left(t\right),f\left(t\right)-\sum_{k}u_{k\left(t\right)}\rangle ,$$
where α is the regularization parameter representing the variance of the white noise, and λ(t) is the Lagrangian multiplier.

The solution of Equation (3) is obtained using the alternate direction method of multipliers (Dragomiretskiy and Zosso, 2014). By iteratively optimizing the saddle points of $u_{k}$, $w_{k}$, and λn + 1 search in Equation (3), the optimal solution of the variational model is obtained, and the center frequency and signal bandwidth of each IMF are continuously updated until the termination conditions are satisfied.

The concept of kurtosis was proposed by the British statistician Karl Pearson in 1887. The term “kurtosis” describes the degree of peaks of the data distribution and is an indicator of the skewness and dispersion of the data distribution. It is widely used in statistics, finance, physics, and other disciplines. Nowadays, kurtosis is also often used in signal time–frequency processing [24,25].

The kurtosis factor K is defined as follows:(4)$$\mathrm{KU}=\frac{1}{N}\sum_{i=1}^{N}{\left(\frac{x\left(i\right)-\bar{x}}{\sigma }\right)}^{4},$$
where $\bar{x}$ is the mean value of signal x(t), and σ is the standard deviation of signal x(t).

When the rolling bearing fails, the amplitude of the bearing vibration signal increases greatly, and the value of the kurtosis factor increases; the larger the amplitude of the fault signal, the more obvious the fault characteristics. IMF components with a kurtosis factor greater than 3 are usually screened out because these components can retain a large number of fault features in the fault signal and abandon the interference of other IMF components to the reconstructed signal (Figure 1).

### 2.2. MFCCs

MFCC was proposed in the 1980s by Davis and Mermelstein [26]. MFCCs are more effective for identification purposes than other parameters. As such, MFCCs have been extensively employed in various speech recognition applications, including instruction recognition, emotion recognition, and person recognition. Moreover, MFCCs are being increasingly used for the status and fault identification of mechanical equipment, such as transformers and UAVs [27,28].

The fundamental concept of MFCC refers to a cepstral coefficient calculated in the Mel-scale frequency domain. MFCC is a parameter that corresponds to the energy of the audio signal in different frequency ranges within the cepstrum. By capturing both the low-frequency envelope and high-frequency detail information, MFCCs can effectively reflect the acoustic characteristics of the signal. The Mel-scale frequency domain simulates the human ear’s perception of audio frequencies and is defined as follows:(5)$$M(f)=2595\;\mathrm{lg}(1+\frac{f}{700}),$$
where M is the Mel frequency, and f is the frequency.

### 2.3. CNNs

CNNs are feed-forward neural networks that include convolution calculations. CNNs greatly reduce the number of network parameters through local connections and weight sharing, and have thus been widely used in the field of computer vision [29,30]. Local connections mean that neurons do not need to be connected one-to-one but only need to be connected in groups. The structure of the image and the pixels in the adjacent area are more closely connected; thus, a single neuron only needs to perceive the local area of the image and then combine the local features of the image obtained by the lower layers in the higher layer. Weight sharing means that each channel uses the same convolution kernel to deconvolute the image at all positions [31,32,33]. CNNs have many structures, but as shown in the LeNet-5 network in Figure 2, CNNs typically include convolutional layers, pooling layers, fully connected layers, and activation functions [34].

In the training process, the loss function, which serves as a metric for evaluating the discrepancy between the model’s predictions and the ground truth, is optimized. This loss function is typically specific to the given task and can take various forms. The cross-entropy function (Equation (6)) is a common example of loss functions used for classification tasks with mutually exclusive classes [19]:(6)$$\mathrm{Loss}=-\frac{1}{M}\sum_{n=1}^{N}\sum_{m=1}^{M}y_{\mathrm{mn}}\ln\hat{y_{\mathrm{mn}}},$$
where M is the number of observations, N is the number of classes, $\hat{y_{\mathrm{mn}}}$ is the network output for the m-th observation and n-th class, and y_mn_ is the ground truth for the m-th observation and n-th class.

### 2.4. EMA-RepVGG

In 2021, Ding [22] proposed a CNN architecture known as RepVGG, which is characterized by a simple yet effective design that incorporates VGG-style deep convolutional networks comprising a sequence of 3 × 3 convolutions and rectified linear unit activations. The architecture is further enhanced by the use of structural reparameterization technology, which enables efficient training and inference on GPUs and specialized chips. To achieve superior training results and feature extraction capability, RepVGG employs a multibranching residual structure similar to that of ResNet. Notably, the training and inference stages of the network utilize different architectures, with training emphasizing accuracy and inference prioritizing speed [34]. The RepVGG architecture comprises five stages, with downsampling achieved via stride-2 convolution at the start of each stage. Figure 3 illustrates the first four layers of a specific stage.

In the field of statistics, the moving average (MA) computation method is used for creating a series of averages for various subsets of a complete dataset and is typically applied to time series data for smoothing out short-term fluctuations [35]. Exponential moving average (EMA), a type of MA, assigns greater weight and importance to the most recent data points, which is consistent with the principles of the human auditory system, placing greater emphasis on recent information. In deep learning, optimization algorithms are a crucial component. Traditional optimization algorithms, such as SGD (Stochastic Gradient Descent) [36] and Adam [37], only use the gradient information from the current time step for updating parameters, which can result in non-smooth updates and oscillations in the parameter space. To address this issue, exponential moving average (EMA) can be introduced to smooth out the gradient information. Specifically, EMA calculates the exponential weighted average of gradients, which makes the updating process smoother. In addition, EMA can be used to calculate the average accuracy or average loss in the evaluation of DL models to better evaluate the performance of models.

The EMA for a series signal can be formulated as follows:(7)$$\theta_{EMA,t+1}=\left(1-\lambda \right)\cdot \theta_{EMA,t}+\lambda \cdot \theta_{t},$$
where $\theta_{t}$ is the parameter at time *t*, $\theta_{EMA,t}$ is the network parameter after moving average at time *t*, and λ is the weighted weight value (usually takes a number close to 1, such as 0.9995).

The working environment of most heavy load bearings in industrial applications is challenging, especially in wind turbines and even offshore wind turbines; thus, the most reasonable monitoring method is to deploy the processing system locally. For this purpose, we used EMA-RepVGG for identifying the condition of the load bearings. MFCCs and VMD-kurtosis denoising technology greatly reduce high-frequency signals and increase the proportion of fault signals under low-speed rotations. The structural reparameterization technique adopted by RepVGG greatly reduces the cost of reasoning, which is conducive to remote deployment and the application of specialized inference chips. In addition, because at very low speeds not every segment of the signal contains bearing fault information, the use of EMA can improve the robustness of model prediction and reduce errors caused by data instability. Table 1 reports the set of hyperparameters adopted in this work.

## 3. Experimental Setup and Data Acquisition

To validate the proposed method, a bearing fault simulator and wind turbine spindle bearing were used for measuring healthy and faulty AE signals of the rolling element bearing.

### 3.1. Bearing Fault Simulator Experiment

As shown in Figure 4, the bearing fault simulator comprises a main bench, a main shaft, an induction motor, and a test bearing. The bench is made of welded section steel and is used to support the main shaft and its bearings; it also bears the force under radial and axial compression. The inner race is installed on the main shaft to realize radial and axial positioning. The outer race is prevented from rotating by a stop (not shown) installed on the bench or a radial compression jack. The main shaft is installed on the bench by using support bearings at both ends. The motor reducer installed at one end drives the main shaft to transfer and drive the inner ring of the tested bearing to rotate. The motor reducer adopts a variable frequency drive, which can adjust the speed from 1 to 40 rpm.

Double-row tapered roller bearings were used. The bearing parameters are presented in Table 2.

Four health states were considered: normal, rolling element (RE) damage (Figure 5a), outer race (OR) damage, and multirolling element (MRE) damage (Figure 5b). Faults (diameter: 5 mm, depth: 0.5 mm) were mechanically machined on the race that was most loaded in the case of an axial load. The bearings were dismounted, and faults were drilled on the race of interest by using a solid carbide drill with a diameter of 2 mm. Although the produced faults represent localized defects in rolling bearings, the AE data extracted cannot provide a comprehensive scenario of defects detectable in rolling bearings.

Four load cases at four shaft speeds were analyzed (Table 3 and Table 4). Next, 200 signals were extracted for each health state, totaling 800 signals. The AE signals were acquired using the Express-8 AE acquisition system (American Physical Acoustics Company) and sampled at 25,000 Hz. Each round of acquisition lasted 10 s.

Non-overlapping chunks of 1 s duration were extracted from the AE signals to construct the dataset (Table 5). Each signal yielded 10 chunks, resulting in a balanced dataset of 8000 samples across the four classes: normal, RE, OR, and MRE. The experimental results provided labeled data for use in supervised learning. The dataset size was sufficient for use in large DL architectures.

The proposed method’s applicability was analyzed using a randomly split dataset. The data were divided using a common DL splitting method, with 80% used for training the RepVGG model, 10% used for validation, and the remaining 10% used for testing. Details regarding the data split are presented in Table 6.

### 3.2. Wind Turbine Spindle Bearing on-Line Experiment

The experimental process is illustrated in Figure 6. The AE signals of the main bearing of a variable-speed constant-frequency doubly fed wind turbine unit were monitored. The main components of the transmission system were the main shaft, bearing, gearbox, coupling, generator, and main frame. To avoid the interference of the gearbox noise signal, the sensors were placed on the front bearing (near the hub side). Four DP15I AE sensors with a frequency of 60–400 kHz were used on each wind turbine, arranged on the inner wall of the main bearing counterclockwise. The diameter of the inner wall of the main bearing was approximately 50 cm, and the arc distance between each sensor was approximately 39 cm.

For the on-line experiment, the 24 h real-time monitoring method was adopted to conduct 24 h long-term monitoring on the main bearings of the three wind turbines. The bearings were classified according to the health status as healthy and faulty based on the characteristics of the distribution law of AE signals during the rotation of the main bearings of the fans. The waveform flow technology was used to collect and process the waveform flow data (duration: 60 s) with 24 h as the cycle, 1 h as the time interval, and 1 min of data in each hour as the time length.

## 4. Experimental Results and Discussion

We studied the ability of EMA-RepVGG, especially RepVGG A0, to perform bearing fault diagnosis. To validate the performance of this method, AE signals from the bearing fault simulator and wind turbine spindle bearing on-line experiment were acquired using the testbed. As discussed in Section 2.4, the MEA-RepVGG convolutional architecture can achieve feature extraction and fault diagnosis of extreme-low-speed heavy load bearings. Signal processing examples of normalized AE signals for the normal, RE, OR, and MRE damage states are shown in Figure 7, Figure 8, Figure 9 and Figure 10, where (a) is the original signal, (b) is the signal after denoising, and (c) are MFCCs, respectively. The corresponding feature map output from the EMA-RepVGG feature extractor is shown in Figure 10a–c. The information contained in the multifaceted MFCC spectrograms was translated and synthesized in a low-dimensional feature space through feature embedding. The classifier discerns classes by learning the differences between feature embeddings.

The model was fine-tuned using the hyperparameters listed in Table 1. The training time was 1611 s on a standard laptop with Nvidia 1650Ti GPU. The model was implemented in Python 3.7 and PyTorch. The feature map in stage 4 of RepVGG with the three damage classes are shown in Figure 11, from which it can be seen that the model extracts fault information. The loss functions of the proposed model and RepVGG during training are shown in Figure 12. The use of EMA prevented model fluctuations caused by data containing noise and large data gaps and model fluctuations near the optimal solution. The accuracies and reasoning times presented in Table 7 indicate the applicability of the diagnosis model to the test data. The reasoning time of the proposed model was much shorter than that of other models under the same accuracy rate. The complete confusion matrices obtained from the laboratory and wind turbine are shown in Figure 13. The diagnostic accuracy of the proposed model on the bearing of the industrial wind turbine was 86.25%. Thus, the results demonstrate that the classifier yields a high precision and recall rate, as reported in Table 8.

Furthermore, the proposed model was compared with ResNet-50 [38], EfficientNet V1 [39], EMA-RepVGG with FFT, and EMA-RepVGG without VMD-kurtosis denoising. The accuracies obtained using different models are presented in Table 7, and the precision and recall for different classes are listed in Table 8. Although ResNet and RepVGG use a similar residual technique, the use of the structural reparameterization technique greatly increased the reasoning speed of RepVGG. The EfficientNet V1 model exhibited satisfactory accuracies and reduced training times; however, some amount of overfitting was observed. EMA-RepVGG with FFT yielded lower accuracies than the proposed model because MFCCs can better reflect the low-frequency part of the signal; thus, RepVGG with MFCCs is more suitable for low-speed industrial equipment. EMA-RepVGG without VMD-kurtosis denoising yielded a lower accuracy than the proposed model under the same training cycle; although its overall training duration and reasoning duration were less than that of the proposed model, subsequent experiments revealed that it took more time to reach the same accuracy by improving the number of training cycles alone without preprocessing and denoising, which also demonstrates that for a certain understanding of signal fault information, proper pretreatment can optimize the fault diagnosis process.

However, this poses a problem in the interpretation of diagnosis results; the characteristic graph of each convolution kernel in the convolution layer has no clear physical interpretation. In contrast to traditional signal processing tools, where some parameters (e.g., kurtosis, crest factor, and ball passing frequencies) have a physical meaning, the user does not know what the features represent for data-driven fault diagnosis, even though they may work perfectly. Thus, estimating the feature variability with respect to the changes in the input signals is challenging. Furthermore, the development of appropriate tools for interpretability is crucial to accurately visualize the alignment of domains in DL [18].

## 5. Conclusions

An extreme-low-speed heavy load bearing fault diagnosis model was proposed using AE signals and DL. VMD-kurtosis denoising was used for data preprocessing, and MFCCs were used as the model input. An improved model, EMA-RepVGG, was then established for feature extraction and classification. Experimental data were obtained from the experimental set-up at the NEPU and wind turbine main shaft bearing of the wind turbine. The experiment involved four bearing states, four rotational speeds, and four loads, with a total of 8000 signals. The wind turbine spindle bearing experiment involved two wind turbines with different conditions in the same wind farm. AE signals were classified with an accuracy of 98.57%. The training time was 1611 s. Based on the results, the following conclusions can be drawn:AE technology can be employed for the signal acquisition of low-speed heavy load bearings.DL CNNs can be used for industrial condition monitoring.The EMA-RepVGG model is an industrial application model suitable for remote deployment and embedded development.MFCCs focus on low-frequency signals in AE signals, which is beneficial for low-speed bearing failure detection.Denoising the model input aids in the optimization of the training process under specific circumstances.

## Figures and Tables

**Figure 1 sensors-23-03541-f001:**
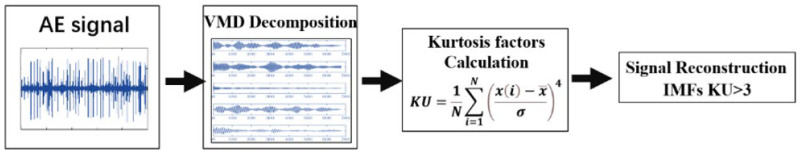
Flowchart of the VMD-kurtosis denoising method. First, VMD decomposition is applied to the acoustic emission (AE) signal to convert the signal into eight IMFs. Next, the kurtosis factor value of each IMF is calculated. Finally, the IMF with kurtosis factor > 3 is reconstructed.

**Figure 2 sensors-23-03541-f002:**
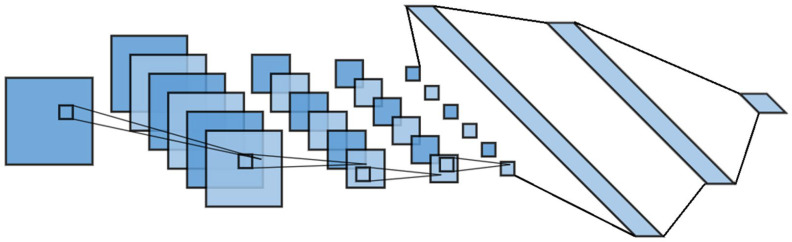
Typical CNN network: LeNet-5.

**Figure 3 sensors-23-03541-f003:**
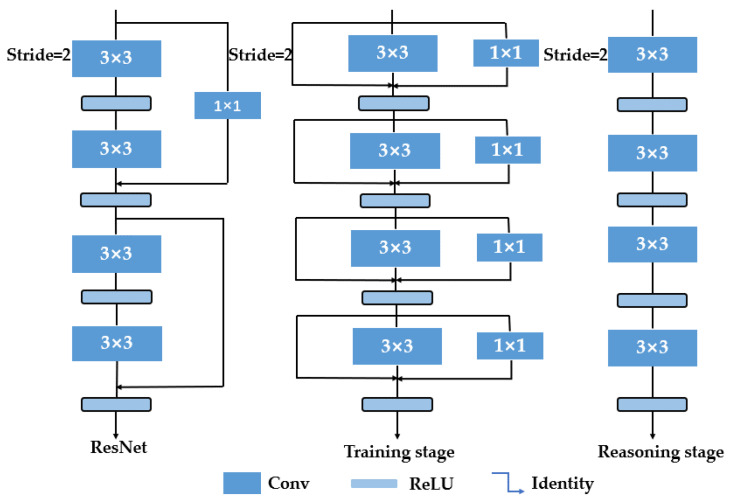
RepVGG network architecture. Left image: residual structure in ResNet. Middle image: training phase, using a similar multibranch residual structure inspired by ResNet. Right image: inference stage, converting all the network layers to Conv 3 × 3 through the reparameterization process. The rectified linear unit is a piecewise linear activation function that outputs the input directly if it is positive and zero otherwise [22].

**Figure 4 sensors-23-03541-f004:**
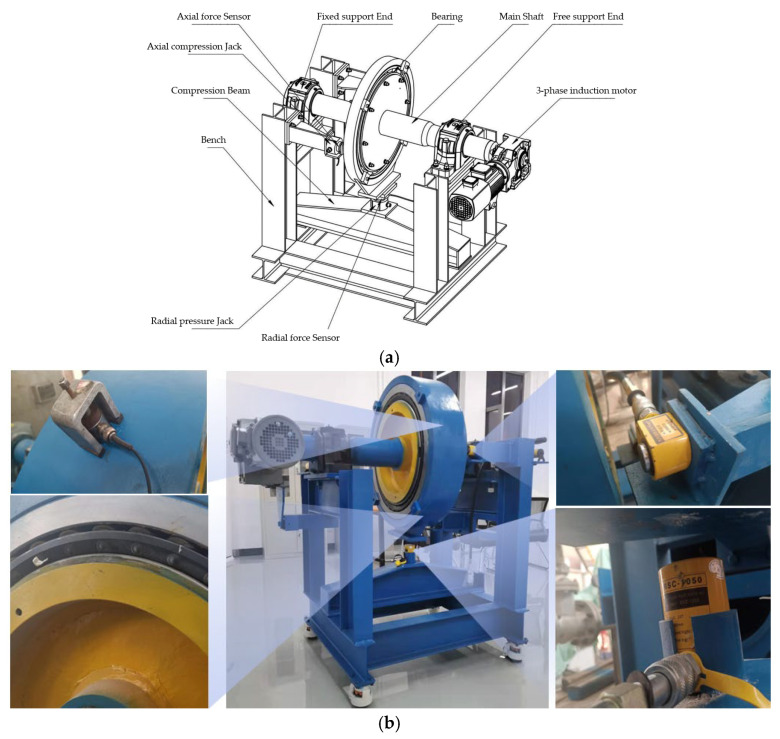
Bearing fault simulator: (**a**) diagram of the bearing fault simulator; (**b**) photographs of the bearing fault simulator.

**Figure 5 sensors-23-03541-f005:**
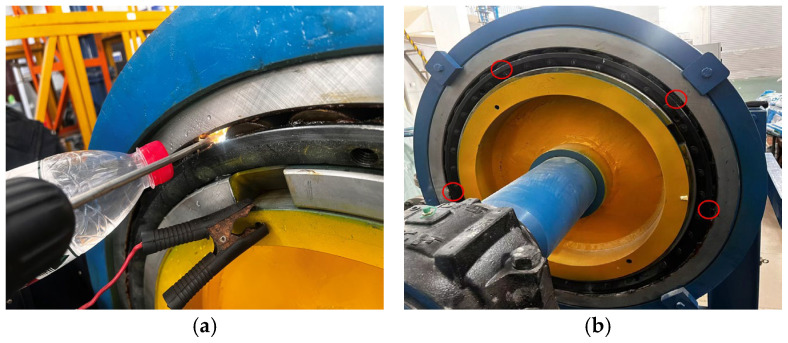
SKF 3519/560/HCYA6: (**a**) rolling element damage with 2 mm diameter and 0–5 mm depth; (**b**) multirolling element damage with 2 mm diameter and 0.5 mm depth.

**Figure 6 sensors-23-03541-f006:**
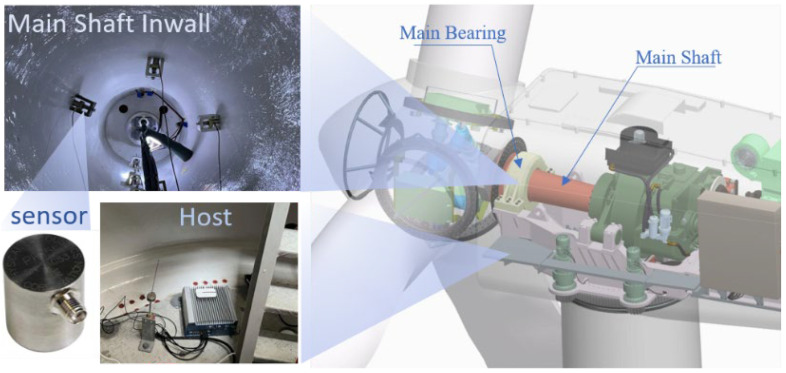
The wind turbine spindle bearing on-line experimental setup.

**Figure 7 sensors-23-03541-f007:**
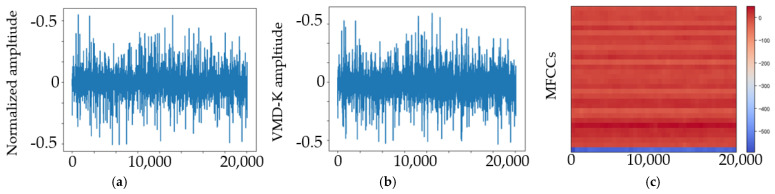
AE signals of the normal health state at 20 rpm and 100 kN radial load: (**a**) original signal; (**b**) signal after denoising; (**c**) MFCCs.

**Figure 8 sensors-23-03541-f008:**
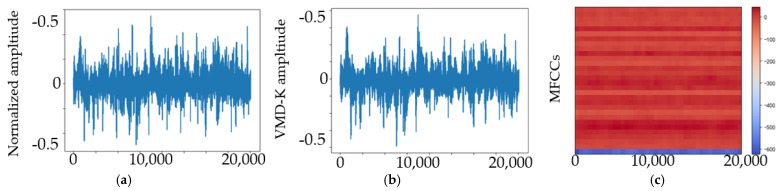
AE signals of the RE damage state at 20 rpm and 100 kN radial load: (**a**) original signal; (**b**) signal after denoising; (**c**) MFCCs.

**Figure 9 sensors-23-03541-f009:**
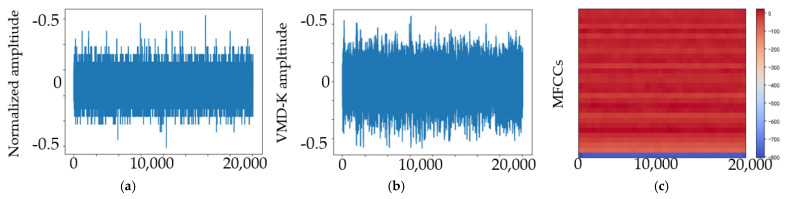
AE signals of the OR damage state at 20 rpm and 100 kN radial load: (**a**) original signal; (**b**) signal after denoising; (**c**) MFCCs.

**Figure 10 sensors-23-03541-f010:**
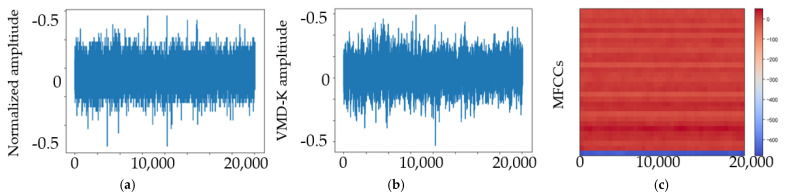
AE signals of the MRE damage state at 20 rpm and 100 kN radial load: (**a**) original signal; (**b**) signal after denoising; (**c**) MFCCs.

**Figure 11 sensors-23-03541-f011:**
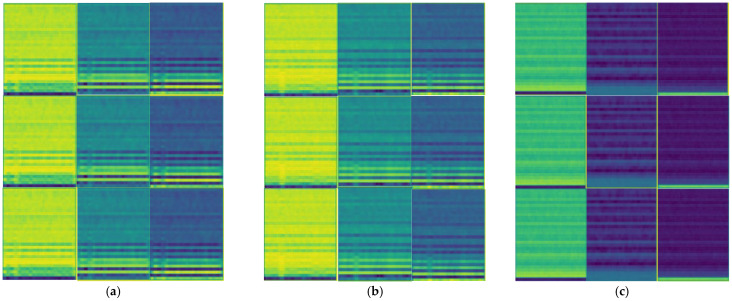
Feature map in stage 4 of RepVGG: (**a**) RE damage; (**b**) OR damage; (**c**) MRE damage.

**Figure 12 sensors-23-03541-f012:**
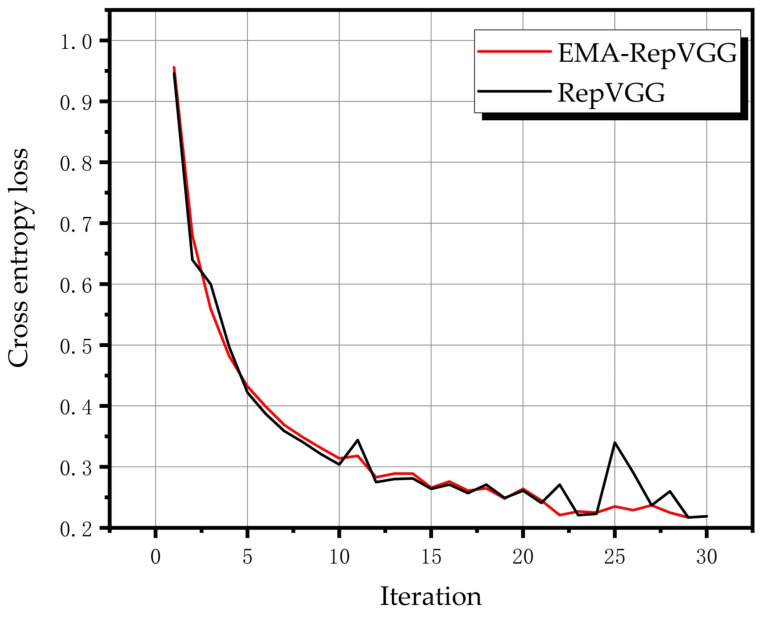
Loss functions.

**Figure 13 sensors-23-03541-f013:**
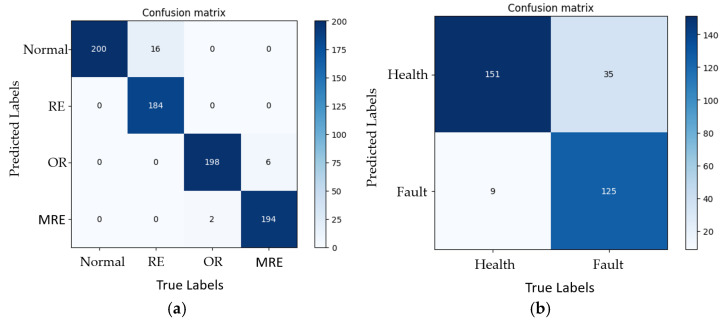
Test confusion matrix: (**a**) bearing fault simulator; (**b**) wind turbine spindle bearing.

**Table 1 sensors-23-03541-t001:** Hyperparameters for EMA-RepVGG transfer learning.

Hyperparameters	Value
Optimizer	Adam [37]
L2 regularization	1 × 10^−6^
Mini batch size	32
Initial learning rate	5 × 10^−4^
Max epochs	30
λ	0.999

**Table 2 sensors-23-03541-t002:** Bearing parameters of indoor acoustic emission test bearing.

Bearing Outer Diameter D	Bearing Inner Diameter d	Thickness	Number of Rolling Elements	Contact Angle
733.4 mm	509.94 mm	200 mm	2 × 35	45°

**Table 3 sensors-23-03541-t003:** Parameter setting of the AE monitoring system.

Project	Channel	Threshold (dB)	PDT (μs)	HDT (μs)	HLT (μs)
parameter	4	35	1000	2000	2000

PDT (Peak Definition Time), HDT (Hit Definition Time), HLT (Hit Locking Time).

**Table 4 sensors-23-03541-t004:** Test conditions.

	Load Case 1	Load Case 1	Load Case 1	Load Case 1
Radial load (kN)	0	50	100	100
Axial load (kN)	0	0	0	50
Nominal speeds (rpm)	5, 10, 15, 20			

**Table 5 sensors-23-03541-t005:** Signal extraction.

Total acquisition duration (s)	10
Sampling frequency (Hz)	25,000
Chunk length (samples)	25,000
Chunk length (s)	1.0
Number of chunks per signal	10

**Table 6 sensors-23-03541-t006:** Details of the dataset.

Classes	Label	Training Samples (80%)	Validation Samples (10%)	Test Samples (10%)
4	Normal	1600	200	200
	IR	1600	200	200
	OR	1600	200	200
	RE	1600	200	200
	Total	8000	800	800

**Table 7 sensors-23-03541-t007:** Diagnosis accuracy and reasoning time.

Model	Training Accuracy	Test Accuracy	Training Time (s)	Reasoning Time (s)	Peak Memory Allocated (MB)
**EMA-RepVGG with** **MFCCs**	**98.88%**	**97.57%**	**1611**	**9.08**	**305**
EMA-RepVGG with FFT	95.46%	95.05%	1688	9.12	305
EMA-RepVGG without VMD-Kurtosis Denoising	91.48%	92.68%	1679	9.49	305
ResNet [38]	96.44%	96.25%	4640	15.27	477
EfficientNet V1 [39]	90.12%	92.96%	1501	13.99	430

**Table 8 sensors-23-03541-t008:** Precision and recall of diagnosis models.

Model	Label	Precision	Recall	Specificity
**EMA-RepVGG with** **MFCCs**	Normal	93.1%	100%	97.9%
	ER	100%	92.3%	100%
	OR	97.0%	99.2%	99.1%
	MER	99.3%	97.3%	99.8%
EMA-RepVGG with FFT	Normal	90.8%	93.2%	92.1%
	ER	91.7%	91.6%	94.7%
	OR	91.3%	92.7%	93.2%
	MER	93.4%	94.1%	94.1%
EMA-RepVGG without VMD-Kurtosis Denoising	Normal	89.1%	90.8%	92.6%
	ER	92.6%	88.9%	89.7%
	OR	92.4%	89.7%	91.3%
	MER	93.4%	91.7%	93.4%
ResNet 50 [38]	Normal	94.4%	92.0%	98.2%
	ER	100%	94.5%	100%
	OR	99.0%	99.0%	99.7%
	MER	92.1%	99.5%	97.2%
EfficientNet V1 [39]	Normal	91.3%	93.9%	97.0%
	ER	93.5%	90.7%	98.0%
	OR	93.9%	92.5%	98.1%
	MER	93.4%	94.6%	97.5%

## Data Availability

Data can be provided upon request from the correspondence author.

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
