# Peer review of "Extreme-Low-Speed Heavy Load Bearing Fault Diagnosis by Using Improved RepVGG and Acoustic Emission Signals"

_sensors, 2023, doi:10.3390/s23073541_

Round 1

Reviewer 1 Report

In this paper, the authors proposed a bearing fault diagnosis technique based on the use RepVGG deep learning network.  Before introducing the data into the RepVGG network the normalized and noise-reduced bearing signals were first converted into Mel frequency cepstrum coefficients (MFCCs) and the exponential moving average method was used to optimize the model and improve its accuracy.

The paper appears to be well written and clearly presented. It seems well organized and the English language  is clear and correct.  However, it needs some attention and needs to be further improved.

In line 179 Ding proposed: Who is Ding? Please indicate the reference.

In figure 1. What do you mean by signal reconstruction? 

Lines 202 and 203: What is the meaning of SGD and Adam? If it is an abbreviation, it must be specified and references must be added.

In Table 3, Abbreviations PDT, HDT and HLT must be defined.

Line 286: Four AE sensors were used on each wind turbine. Please specify the used sensors and their characteristics.

The sentence between lines 303 and 307 is too long and should be rewritten in shorter sentences.

Line 336 confusion matrixes must be corrected to confusion matrix or confusion matrices.

Lines 355-357:  EMA-RepVGG with FFT yielded lower accuracies than the proposed model because MFCCs can better reflect the low-frequency part of the signal; thus, RepVGG with FFT is more suitable for low-speed industrial equipment. It seems that there is a contradiction in this sentence. Please correct it or justify the conclusion.

Author Response

Point 1: In line 179 Ding proposed: Who is Ding? Please indicate the reference.

Response 1: There are indeed no references mentioned in the article, and I have included them in the new version.

Point 2: In figure 1. What do you mean by signal reconstruction? 

Response 2: Signal reconstruction means reconstructing several reserved IMFs components together to obtain a denoised signal. so no changes have been made in the text.

Point 3: Lines 202 and 203: What is the meaning of SGD and Adam? If it is an abbreviation, it must be specified and references must be added.

Response 3: Both SGD and Adam are commonly used optimizers in deep learning. Adam does not have a full name, and the full name and references of SGD have been modified in the text.

Point 4: In Table 3, Abbreviations PDT, HDT and HLT must be defined.

Response 4: I have defined the above parameters in the article.

Point 5: Line 286: Four AE sensors were used on each wind turbine. Please specify the used sensors and their characteristics.

Response 5: I have indicated the sensor model and parameters in the article.

Point 6: The sentence between lines 303 and 307 is too long and should be rewritten in shorter sentences.

Response 6: I have modified these sentences in the attachment.

Point 7: Line 336 confusion matrixes must be corrected to confusion matrix or confusion matrices.

Response 7: I have changed the expression in the article.

Point 8: Lines 355-357:  EMA-RepVGG with FFT yielded lower accuracies than the proposed model because MFCCs can better reflect the low-frequency part of the signal; thus, RepVGG with FFT is more suitable for low-speed industrial equipment. It seems that there is a contradiction in this sentence. Please correct it or justify the conclusion.

Response 8: It should have been written incorrectly at the time, and I have already changed it in the article.

Reviewer 2 Report

1. In Line 234 "the bearing is installed on the inner race and outer race of the bearing", Please check the entire manuscript for such meaning less sentences.

2. With the solid carbide drill of diameter 2mm, how the fault diameter of 5mm was achieved (Line 251-255)? 

3. What is 2 X 35 under the column number of rolling elements in Table 2?

4. It was mentioned that loss function is shown in Figure 11 (Line 330-331), Please make the necessary correction.

Author Response

Point 1: In Line 234 "the bearing is installed on the inner race and outer race of the bearing", Please check the entire manuscript for such meaning less sentences.

Response 1: I have changed the sentence in the article.

Point 2: With the solid carbide drill of diameter 2mm, how the fault diameter of 5mm was achieved (Line 251-255)? 

Response 2: This can be achieved by first drilling out 2mm of damage at the location, and then polishing around it to 5mm.  So I did not modify this section in the article.

Point 3: What is 2 X 35 under the column number of rolling elements in Table 2?

Response 3: Because the bearing used in this article is a double row tapered roller bearing, it has two rows of rolling elements with 35 in each row, so it is 2 × 35, so no changes have been made in the text.

Point 4:  It was mentioned that loss function is shown in Figure 11 (Line 330-331), Please make the necessary correction.

Response 4: The picture annotation in the text is incorrect and has been changed.
